# Polyamines and Their Metabolism: From the Maintenance of Physiological Homeostasis to the Mediation of Disease

**DOI:** 10.3390/medsci10030038

**Published:** 2022-07-15

**Authors:** Kamyar Zahedi, Sharon Barone, Manoocher Soleimani

**Affiliations:** 1Division of Nephrology, Department of Internal Medicine, University of New Mexico Health Sciences Center, Albuquerque, NM 87131, USA; sbarone@salud.unm.edu (S.B.); msoleimani@salud.unm.edu (M.S.); 2Research Service, New Mexico Veterans Administration Medical Center, Albuquerque, NM 87108, USA

**Keywords:** polyamine, polyamine metabolism, putrescine, spermidine, spermine

## Abstract

The polyamines spermidine and spermine are positively charged aliphatic molecules. They are critical in the regulation of nucleic acid and protein structures, protein synthesis, protein and nucleic acid interactions, oxidative balance, and cell proliferation. Cellular polyamine levels are tightly controlled through their import, export, *de novo* synthesis, and catabolism. Enzymes and enzymatic cascades involved in polyamine metabolism have been well characterized. This knowledge has been used for the development of novel compounds for research and medical applications. Furthermore, studies have shown that disturbances in polyamine levels and their metabolic pathways, as a result of spontaneous mutations in patients, genetic engineering in mice or experimentally induced injuries in rodents, are associated with multiple maladaptive changes. The adverse effects of altered polyamine metabolism have also been demonstrated in *in vitro* models. These observations highlight the important role these molecules and their metabolism play in the maintenance of physiological normalcy and the mediation of injury. This review will attempt to cover the extensive and diverse knowledge of the biological role of polyamines and their metabolism in the maintenance of physiological homeostasis and the mediation of tissue injury.

## 1. Introduction

Polyamines are aliphatic polycations that are present in all cells and play an important role in the mediation of biological functions that are necessary for growth and survival [1,2]. Their discovery dates back to 1678, when spermine was first identified as crystals in cooled human sperm [3]. It was not until 1926 that the chemical structure of spermine (Spm) was elucidated [4]. Polyamines are abundant in all organisms and perform important biological functions that are essential for life [5]. The biologically significant polyamines in mammals (Figure 1A), Spm (N-N’-bisaminopropyl-1,4-diaminobutane) and spermidine (Spd; N-aminopropyl-1,4-diaminobutane), are derived by sequential enzymatic addition of aminopropyl groups to putrescine (Put; 1,4-diaminobutane) and are required for growth and development [5,6]. Intracellular levels of polyamines are regulated through their transport, synthesis, and degradation (Figure 1B). The transporters of polyamines have been identified in mammalian cells and organs [7], while polyamine synthetic and degradative pathways have been elucidated in bacteria, yeast and higher organisms [8]. Disturbances in cellular polyamine levels and metabolism are observed in many injuries and in cancer [5,9,10,11,12]. The adverse effects of elevated polyamine metabolism have also been demonstrated *in vitro* [13,14,15,16]. These observations point to the importance of polyamines and their metabolism in the maintenance of physiological homeostasis and mediation of tissue injury. In this review, we will cover the important aspects of polyamine biology and polyamine metabolism in health and disease.

## 2. Polyamines and Regulation of Their Cellular Content in Mammals

In mammals, Spm and Spd are the naturally occurring polyamines, while Put is the primary diamine (Figure 1A). At physiological pH, these aliphatic cations contain at least two positively charged amino groups and serve a variety of biologically important functions. Because of their important biological role, the cellular content of polyamines is tightly regulated via their import, export, synthesis or degradation [5].

### 2.1. Polyamine Metabolism

Polyamine synthesis is initiated by the Arginase 1 (ARG1) catalyzed conversion of arginine to ornithine followed by the decarboxylation of ornithine by ornithine decarboxylase (ODC), the rate-limiting enzyme in polyamine synthetic pathway, to generate putrescine (Put). The activity of ODC is regulated at multiple levels. The transcription of ODC is regulated by MYC [17], while the stability of the ODC mRNA is regulated through binding of HuR, an RNA binding protein, to AU rich regions in its 3′ untranslated region [18]. Additional regulatory mechanisms of ODC expression that affect the translation of its transcript have been also described. These involve: (1) using alternative upstream open reading frames that reduce the translated ODC levels, and (2) a cap-independent translation mechanism for the synthesis of ODC that uses an alternative ribosomal binding site and is most likely used to maintain polyamine synthesis through G2/M transition phase, the transition between the second growth phase (growth phase after S phase) and mitosis in the cell cycle [19,20]. ODC levels are also post-translationally regulated by antizyme 1, which binds to ODC, prevents its dimerization, and targets it for proteosomal degradation [21]. The sequential addition of aminopropyl groups, derived from decarboxylated S-adenosyl methionine (dcSAM), to Put and Spd in reactions that are catalyzed by spermidine synthase (SRM) and spermine synthase (SMS) generates Spd and Spm, respectively.

The catabolism of polyamines is mediated by their back conversion via spermidine/spermine N^1^-acetyltransferase (SAT1)/acetylpolyamine oxidase (PAOX) cascade. The back conversion of polyamines depends on their acetylation by SAT1 in a reaction that utilizes acetyl coenzyme A (Acetyl-CoA) followed by the oxidation of acetylated polyamines by PAOX. The SAT1-mediated acetylation of polyamines is a rate limiting step in their catabolism, and the expression and activity of SAT1 are regulated at multiple levels. The transcription of *SAT1* mRNA and its induction by polyamines or their analogs is regulated by Nuclear factor erythroid 2-related factor 2 (Nrf-2), which binds to the polyamine response element in the promoter region of the *SAT1* gene, and its co-factor polyamine modulated factor 1 (PMF-1) [22,23,24]. Another important factor in the regulation of *SAT1* gene is a SP1 binding site that is essential for its basal transcription [25]. The SP1 and factors that bind to PRE (Nrf-2 and PMF-1) are important in the increased transcription of *SAT1* gene in response to radiation damage [25]. The activation of P53, a DNA-damage-activated gene, has also been implicated in the upregulation of *SAT1* gene transcription in the process of ferroptosis [26]. Tumor necrosis-α (TNF-α), a known activator of NFκB and p53 [27,28,29], also can induce the expression of SAT1 transcription [30,31]. The cross talk between the various mechanisms of transcriptional regulation of the SAT1 gene remains to be elucidated. Post-transcriptionally, SAT1 expression is regulated via a number of mechanisms including: (1) the unproductive splicing of SAT1 transcript in response to cellular polyamine levels [32]; (2) the polyamine-mediated override of translational repression caused by the usage of false upstream open reading frames, allowing the use of the proper translation start site [33]; and (3) the translational repression by nucleolin binding to a potential stem loop structure in the 5′ region of *SAT1* mRNA coding region that is relieved by elevated cellular polyamine levels and consequent degradation of nucleolin [34]. At the post-translational level, the regulation of SAT1 activity is mediated via its dimerization to form the active enzyme [35]. The SAT1/PAOX cascade generates H_2_O_2_, 3-acetylaminopropanal, and acrolein that is generated through the spontaneous deamination of 3-acetylaminopropanal. The generation of these compounds may account for the adverse effect of increased SAT1 expression in cultured cells [14,15].

The oxidation of Spm by spermine oxidase (SMOX), a highly inducible enzyme, is another major mode of polyamine catabolism (Figure 1B). The expression of SMOX is elevated in response to tissue injury and inflammation [36,37,38,39,40]. In injured tissue, increased oxidation of Spm by SMOX generates cytotoxic H_2_O_2_, 3-aminopropanal, and acrolein molecules, which can further exacerbate the severity of tissue damage [5,9,36,39,41]. The expression of SMOX can be induced by treatment with polyamine analogs, as well as pro-inflammatory cytokines such as interleukin-6 or TNF-α [42]. Tissue culture studies indicate that the increased expression of SMOX in response to treatment with the Spm analog, CPENSpm, is dependent on *de novo* synthesis and increased stability of the SMOX transcript [43]. Examination of the SMOX gene promoter region indicates that its polyamine response elements are active in constructs spanning the residues −1117 to −55 of its 5′ untranslated region [43]. These results also suggest that there are multiple responsive elements present in this region that regulate the basal activity of the gene and its responsiveness to polyamines. An additional layer of regulation consists of miR-124 modulation of SMOX expression through its interaction with the 3′ untranslated region of the SMOX mRNA [44]. *In vitro* studies indicate that increased expression of SAT1 in cultured cells enhances the expression of the SMOX mRNA. They further demonstrate that the enhanced expression of the SMOX transcript in response to SAT1 induction is dependent on the activity SAT1, suggesting the existence of crosstalk between SAT1 expression and activity and the expression of SMOX [45].

### 2.2. Polyamine Transport

Polyamine transport is important in the regulation of cellular polyamine levels [7]. The polyamine transporters identified thus far include members of solute carrier (SLC) and p-type ATPase families. Multiple members of the organic cation transporter (OTC/SLC22A) family, such as SLC22A1-3, have been implicated in the transport of Put and Agmatine [46,47,48]. The role of SLC3A2 as a transporter that favors Put and acetylated polyamines has also been well characterized [49,50]. Another member of the SLC family, SLC18B1, that functions as a vesicular transporter was also identified as a transporter of Spd and Spm in neurons and astrocytes [51,52]. Recent studies demonstrated the role of ATP13A3, a P5B-ATPase, which is linked to pulmonary arterial hypertension, in the cellular transport of polyamines in CHO and pancreatic duct carcinoma cells [53,54]. A member of the same family of transporters, ATP13A2, that is associated with an atypical form of juvenile onset Parkinson disease is also implicated in the lysosomal export of polyamine to the cytosol [55]. 

## 3. Biological Functions of Polyamines and Their Potential Physiological Role in Health and Disease

Spm and Spd are polycations that are ubiquitous in all mammalian cells and perform a variety of biologically significant functions [1,2]. Because of their positive charge at physiological pH, the majority of the cellular functions attributed to polyamines are due to their interactions with negatively charged biomolecules (e.g., phosphate back bone of nucleic acids, negatively charged amino acid residues in proteins, and negatively charged membrane phospholipids) [6,56].

### 3.1. Interaction of Polyamines and Nucleic Acids

In mammalian cells, the highest proportion of polyamines is found in association with RNA and DNA, while only about 10% are found as free molecules [57]. Polyamine binding stabilizes transfer- and mRNA molecules and enhances the translation of the latter [6,58,59]. Interaction of polyamines with antizyme mRNA is required for the +1 programmed ribosomal frameshift that leads to an increased production of antizyme and proteasomal degradation of ODC, the rate-limiting enzyme in polyamine synthesis [60]. Studies examining the effect of polyamine on DNA structure indicated a B to Z transition upon incubation with polyamines (Spm and Spd), as well as Put. The structural transition was dependent on the concentration of polyamines and Put, with the former being more effective than Put in the induction of B to Z conformational transition. Furthermore, the long-term incubation with Spm enhanced the aggregation of DNA [61]. The conformational transition of DNA from B to Z may also occur in response to polyamine/diamine complexes known as nuclear aggregates of polyamines (NAP) [62]. This structural transition may be important in the development of anti-Z-DNA antibodies in autoimmune diseases such as systemic lupus erythematosus (SLE) and rheumatoid arthritis [63]. NAPs also have high affinity for G-C rich Alu elements [63,64], which are found in primates and through their transposition can have deleterious genetic effects [65,66]. Polyamines in mM concentration can destabilize the telomeric G-quadruplex structures [67]. With higher polyamine content in actively proliferating cells, the polyamine-mediated destabilization of the G-quadruplex may be important in the process of carcinogenesis [67].

### 3.2. Antioxidant Role of Polyamines

Polyamines can not only directly regulate DNA structure and function [61,63] but can also act as free-radical scavengers or modulators of the Fenton reaction that protect nucleic acids and cells against oxidative damage [68,69,70]. Studies in cell-free systems indicate that the free radical scavenging activity of polyamines is dependent on the concentration of transition metals (e.g., Spm is protective in the presence of low concentrations and can induce DNA strand break in the presence of high concentrations of transition metals) [71]. Additionally, free-radical scavenging activity of polyamines is primarily directed against hydroxyl radicals formed by Fenton-like reactions and against singlet oxygen; however, polyamines are not effective scavengers of super oxide radicals [72]. The free radical scavenging function of Spm has been confirmed in yeast cells and Gy11 cultured mouse embryonic fibroblast cells that lack SMS [73,74]. While GY11 fibroblasts were highly susceptible to H_2_O_2_, normalization of their intracellular polyamine levels protected them against H_2_O_2_-induced injury, thereby confirming the antioxidant role of polyamines [74].

### 3.3. Interaction with Proteins

As positively charged molecules, polyamines interact with a variety of proteins and affect the structure, function, interaction, and localization of these proteins [75,76,77]. While these reactions are charge-dependent, many are further stabilized through enzymatic polyamination of targeted residues [77,78,79]. The predominant enzymatic reaction that drives the polyamination of proteins is catalyzed by transglutaminases [77]. The transglutaminase-mediated polyamination of phospholipase A2 increases its enzymatic activity by nearly threefold compared to its non-polyaminated counterpart [78]. Transglutaminase-mediated polyamination of Tau, a microtubule-associated protein that is found in amyloid deposits of Alzheimer disease, prevents its proteolysis by calpain and may play a role in plaque formation [80]. Spd and Spm were also shown to promote the aggregation and fibril formation by α-synuclein, with Spm being more efficient than Spd as a driver of this reaction [75,81,82,83]. The addition of polyamines increased the rate of nucleation and monomer addition in α-synuclein aggregates [81]. Polyamination of α-synuclein was also demonstrated in a mouse model of combined SMOX and SAT1 deficiency [84]. These animals develop severe cerebellar damage and ataxia [84]. The cerebellar damage is associated with increased expression of transglutaminase 2; as well as expression, polyamination, and aggregation of α-synuclein [84]. Treatment of *Smox/Sat*-1 double knock-out mice (*Smox**/Sat1*-dKO) with transglutaminase inhibitors decreased the expression and aggregation of polyamines, reduced the severity of cerebellar damage, and significantly delayed the onset of ataxia [84]. 

Depending on the applied dosage, intracerebroventricular administration of polyamines leads to hypothermia, sedation, and convulsions [85]. The function of polyamines in neurotransmission has been demonstrated in single neurons studies, where these molecules had either depressive or excitatory functions depending on the type of the neuron [86,87]. Such observations suggest that polyamines are important regulators of neuronal function and their activity may vary based on their source (intracellular vs. extracellular), nature of the channel molecules, auxiliary factors associated with the channel protein, and more specifically, the target cell type [86,87]. These studies led to the discovery of multiple channels that are affected by polyamines including ionotropic glutamate receptors and potassium channels [86,88]. Polyamines have been established as modulators of glutamate receptors, such as calcium dependent AMPA and kainite receptors [86,89,90]. The regulation of these channels is mediated through blocking their ion channel activity, in addition to their regulation by auxiliary molecules that interact with the channel proteins [86]. Activity of NMDA receptors that contain the GluN2B subunit are also regulated by polyamines, where polyamines are thought to stabilize the receptor in its active form [91]. Polyamines also control the activity of inward rectifier potassium channels, which is likely mediated through steric blocking of their ion channel pore [92,93].

### 3.4. Spd and Hypusination of Eukaryotic Translation Elongation Factor 5A (eIF5A)

Hypusine is an amino acid derived through modification of a lysine residue. It is found only in the lysine 50 residue of eIF5A [94,95,96]. Hypusination of eIF5A is initiated when the aminobutyl group of Spd is transferred, in an NAD+ dependent reaction catalyzed by deoxyhupusine synthase (DHPS), to the ε-amino group of its lysine residue 50 to generate a deoxyhypusine residue [97]. The hypusination process of eIF5A is then completed in a reaction catalyzed by deoxyhypusine hydroxylase that adds a hydroxyl group to carbon 9 of the deoxyhypusine residue, forming hypusine residue [97]. Although originally thought to be involved in the ease of processing proline residues during translation [98,99], there are now other functions and physiological roles attributed to eIF5A [96]. The depletion of eIF5A can stall protein synthesis and lead to the formation of stress granules (translationally stalled messenger ribonucleoprotein complexes) and the onset of endoplasmic reticulum stress [100,101]. The targeting of eIF5A can also impart tolerance to ischemia and endotoxic acute kidney injuries, enhance metabolic adaptation in diabetes, alter macrophage and immune cell functions, and affect embryonic and neuronal development [96,102,103,104,105]. The physiological importance of hypusination of eIF5A was also observed in heterozygote *Dhps*-KO mice and in individuals with mutations in the *DHPS* gene [106,107]. Examination of the mutant proteins *in vitro* revealed that the mutated DHPS proteins have either reduced or no enzymatic activity [106]. The reduced or absent activity of DHPS enzyme in these individuals led to neurodevelopmental phenotypes including seizures and delayed development [106]. 

### 3.5. Polyamines: Toxic or Panacea?

The effects of exogenous administration of Spm and Spd have been the subject of many *in vitro* and *in vivo* studies. While a number of studies showed the administration of Spm, and to a lesser extent Spd, to be toxic others suggest that exogenous polyamine administration may be beneficial [11,108,109,110]. The identification of bovine serum amine oxidase-like enzymes and other extracellular amine oxidases that target diamines and polyamines is now considered as the likely mechanism behind the toxicity of polyamines [111,112]. The degradation of polyamines by extracellular amino oxidases generates H_2_O_2_ and reactive aldehydes, including acrolein that is spontaneously formed via conversion of aldehydes generated when polyamines are oxidized. These are thought to be the instigating factors that mediate the toxicity of exogenous polyamines [111,112,113,114]. The ubiquitous presence of extracellular amino oxidases and the importance of their activity in the mediation of tissue injury are supported by studies that show increased polyamine oxidation in the plasma of patients after cerebral stroke, in patients with kidney failure, and in the vitreous humor of patients with diabetic retinopathy [115,116,117,118]. Furthermore, in a rat model of cerebral ischemic reperfusion injury (IRI), treatment with aminoguanidine, an amino oxidase inhibitor, was found to reduce the production of 3-aminopropanal and to protect against neuronal and glial cell death [119]. Additional studies confirmed the role of aminoaldehydes in the mediation of tissue damage in this model of cerebral IRI [120]. The results presented above point to the potential role of extracellular amino oxidases and their cytotoxic products as mediators of injury in many studies where toxicity of exogenous polyamines was documented.

The protective role of polyamines and their derivatives has also been demonstrated in a number of studies [11,108,109,121]. Spm was shown to reduce the lesion size in a model of cerebral hypoxia-ischemia in rat pups, where neuroprotection imparted by Spm was due to increased generation of neuroprotective nitric oxide and reduced mitochondrial damage [108]. Similarly, exogenously administered Spm was shown to protect against manganese-induced neurodegeneration in α-synuclein-expressing *Caenorhabditis elegans* (*C. elegans*) [122]. Exogenously administered Spm, as well as elevated intracellular polyamine levels, also protected the α-synuclein-expressing SK-MEL-28 cell line against manganese-induced injury [122]. The protective role of Spm has also been demonstrated in cardiomyocytes exposed to *in vitro* conditions that mimic ischemic IRI [123]. This protection was conferred through inhibition of the mTOR pathway and the enhancement of autophagic flux [123].

The protective effects of Spd and its derivatives are also extensively studied [11]. Dietary Spd supplementation suppressed the age dependent loss of neuronal plasticity and impairment of memory formation in *Drosophila melanogaster* (*D. melanogaster*) [124]. Dietary provision of Spd also increased the longevity of yeast, *C. elegans* and *D. melanogaster* [125]. Similar to studies conducted with Spm, dietary Spd also reduced the neurotoxicity of α-synuclein in *C. elegans* and *D. melanogaster* [126]. The above, as well as other neuroprotective activities of Spd including the preservation of locomotor activity and promotion of stress resistance in *D. melanogaster*, were shown to be mediated through the induction of autophagy [125,126,127,128]. In mice, provision of Spd in their diet provided protection against cardiac damage caused by age. This was achieved through protecting the cardimoyocytes against injury via the induction of autophagy, enhanced mitochondrial respiration, and improved cardiomyocyte function [129]. Oral administration of Spd prior to induction of renal IRI in mice also reduced the oxidative injury to DNA, the activation of poly(ADP-ribose) polymerase1 (PARP1), and the severity of renal injury [130]. In Staurosporine-induced cell injury, Spd was shown to protect PC12 and cortical neurons against damage via prevention of Becli1 cleavage and induction of autophagy [131]. In autoimmune encephalomyelitis, a model of multiple sclerosis, intraperitoneal administration of Spd was shown to reduce the severity of injury by modifying the macrophage response toward an inhibitory phenotype, which was associated with reduced expression of inflammatory mediators and enhanced expression of Arg1 [132]. A catabolically stable functional mimetic of Spd, α-methylspermidine, exhibited protection against CCl_4_-induced liver and pancreatic injury in spermidine/spermine N^1^-acetyltransferase transgenic rats [133].

## 4. Polyamine Metabolism and Its Role in Health and Disease

The balance of polyamine synthesis and catabolism is critical to the regulation of cellular polyamine levels. Alterations in polyamine metabolism *in vitro*, in genetically engineered animals and in patients due to spontaneous mutations in polyamine pathway genes, can have severe consequences [5,9,12].

### 4.1. Effect of Dysregulation of Polyamine Synthetic Pathway

The *de novo* synthesis of polyamines is catalyzed by ODC, the rate limiting enzyme responsible for the generation of Put, followed by sequential addition of aminopropyl groups from dcSAM to Put by SRM and SMS to generate Spd and Spm, respectively (Figure 1B).

The importance of ODC in the regulation of cell growth, differentiation, and function has been well established [134,135]. The physiological importance of ODC was further established in knock-out mice, where the homozygote ablation of its alleles was shown to be lethal and to have adverse developmental effects in heterozygote mice [136,137]. ODC expression and regulation is altered in response to injuries and in cancer [16,17,138,139]. ODC is a transcriptional target of Myc [17], and its expression is increased in Myc-amplified tumors [140,141]. The adverse role of ODC in tumorigenesis is supported by the observation that its over-expression promotes skin tumorigenesis, and that ODC over-expressing cells when injected into immunocompromised mice can enhance tumor formation [140,142,143].

Genetic polymorphisms in ODC genes are also associated with a number of cancers including neuroblastomas, as well as gastric, colorectal, breast and prostate cancers [144,145,146,147,148]. Yet, maintaining physiological levels of ODC is vital for cell growth and function, as demonstrated through *in vitro* studies [134,135]. Intestinal epithelial cells displayed reduced growth resulting from inhibition of ODC and subsequent polyamine depletion leading to activation of the Transforming growth factor-β mediated SMAD signaling pathway [149,150,151]. Using keratinocytes from normal and ODC over-expressing mice, it was demonstrated that the over-expression of ODC was associated with the induction of DNA damage, activation of the DNA damage checkpoint protein kinase, ataxia telangiectasia mutated (ATM), and cell cycle arrest [16]. DNA damage in this cell model was attributed to the induction of the polyamine catabolic enzyme SMOX and the toxic products resulting from its enzymatic activity [16].

The expression and activity of ODC increases following acute ischemic injuries in the targeted organs [138,139]. The role of ODC in the mediation of tissue injury was most closely studied in cerebral IRI in rats. ODC-expressing transgenic rats were protected against cerebral ischemia, while the inhibition or knockdown of ODC exacerbated the severity of injury in multiple studies indicating a protective role for ODC [152,153,154,155]. These results suggest that ODC plays a protective role in cerebral IRI.

Although the physiological importance of polyamine metabolism has been well documented, it was not until recently that mutations in polyamine biosynthetic pathway enzymes, ODC and SMS, were identified. Description of multiple individuals with gain of function ODC mutations revealed that such mutations lead to macrosomia, macrocephaly, delayed visual maturation, sensorineural hearing loss, and ectodermal abnormalities (e.g., alopecia) along with other presentations [156,157]. In a single case of ODC gain of function mutation, eflornithine normalized the polyamine levels while improving alopecia and neurological symptoms [158]. A loss of function mutation of ODC (gly84Arg) was also recently described in South Asian individuals [137]. This missense variant reduces the activity of ODC by nearly 60% and is associated with psychiatric and neurological traits [137].

In male mice, the mutations in SMS and the absence of tissue Spm leads to neurological abnormalities, growth defects and a shortened lifespan [159]. In humans, Snyder–Robinson Syndrome (SRS) is a recessive X-linked disorder that is attributed to loss of function mutations in the SMS gene [160,161]. SRS is characterized by intellectual disability, skeletal and muscular defects, as well as seizures [161]. Similarly, lymphoblast cell lines derived from individuals with SRS present with significant derangements in cellular polyamine levels. Levels of Put and Spm are considerably reduced, while Spd levels are elevated. Further examination of these cells revealed increased content of tissue transglutaminase 2, an enzyme associated with neurodegenerative diseases [162]. The interplay of increased cellular Spd content and elevated TGM2 levels in the mediation of tissue abnormalities in SRS remain to be elucidated.

### 4.2. Polyamine Catabolism in Health and Disease

Catabolism of polyamines through their back conversion via the SAT1/PAOX cascade and oxidation of Spm by SMOX is important in maintaining homeostasis of these important molecules [12]. Polyamine back conversion/oxidation and Spm oxidation while contributing to the maintenance of physiological levels of polyamines also generate H_2_O_2_ and reactive aldehydes that promote tissue damage in conditions where polyamine catabolism is elevated.

#### 4.2.1. Polyamine Catabolism and Tissue Damage

The expression of SAT1 and SMOX increases in the brain, cardiac, liver, kidney, and GI tract in response to a variety of injuries [9,40,45,163,164,165,166]. The induction of polyamine catabolism in the liver after renal IRI or partial nephrectomy suggests that heightened polyamine catabolism contributes to tissue damage in organs that are not the direct target of the injurious insult [167]. In the kidney and liver, injuries caused by toxic compounds (cisplatin or CCl_4_), sepsis (bacterial lipopolysaccharide), or IRI enhance the expression of both SAT1 and SMOX [167]. The ablation of SAT1, as well as its inhibition, reduced the severity of tissue damage in renal IRI [139,168]. In cisplatin-induced acute kidney injury, the deficiency or inhibition of polyamine catabolic enzymes, as well as the neutralization of their toxic by-products (e.g., reactive aldehydes and H_2_O_2_), reduces the severity of acute tubular damage [39]. The protective effect of the ablation or inhibition of polyamine catabolism was revealed to result from the reduction in oxidative injury, modulation of the innate immune response, and down-regulation of endoplasmic reticulum stress/unfolded protein response [39,164,165,168].

The expression of SAT1 and SMOX is increased in the brain after IRI and traumatic injury [40]. The inhibition of polyamine oxidation by MDL72525, as well as the neutralization of reactive aldehydes generated as a result of polyamine oxidation by N-2-mercaptopropionyl glycine (N-2-MPG), reduce the severity of ischemic damage to the brain [120,123]. The importance of SMOX in the mediation of brain injury is further demonstrated in SMOX transgenic mice that show hypersensitivity to kainic acid, pentylentetrazole-induced seizures, enhanced neurotoxicity, and reactive astrocytosis [169,170,171,172].

Increased expression and activity of SMOX has been documented in response to bacterial endotoxin and in bacterial infections [36,38,164]. In *Helicobacter pylori* infections, the expression of SMOX is associated with chronic inflammation and increased risk of developing gastric cancer [173]. Biopsies of *H. pylori* infected individuals with gastritis demonstrated that elevated levels of SMOX expression in the samples were associated with increased risk of developing gastric cancer [38,174]. The potential correlation between SMOX and gastric cancer was further established in a Mongolian gerbil model of *H. pylori*. In these studies, the expression of SMOX, induction of DNA damage, and development of cancerous lesions was associated with a strain of *H. pylori* that was more likely to lead to gastric cancer vs. that of the low-cancer-risk strain [174]. The role of SMOX in tumorigenesis was demonstrated in a model of *Bacteroides fragilis (B. fragilis)*-induced colon cancer [36]. In these studies, *B. fragilis* was shown to induce the expression of SMOX and lead to DNA damage in infected culture cells. Furthermore, *B. fragilis* infection in wildtype mice led to increased expression of SMOX, while its inhibition reduced the severity of colitis [36]. Moreover, the induction of colon tumorigenesis by *B. fragilis* was reduced when animals were treated with MDL72527, an inhibitor of FAD-dependent polyamine oxidases such as PAOX and SMOX [36].

#### 4.2.2. Polyamine Catabolism and Cell Injury

Enhanced catabolism of polyamines causes cell injury through the disruption of polyamine homeostasis and generation of toxic metabolites. Treatment with polyamine analogs such as *N*^1^,*N*^11^-bis(ethyl)norspermine (BENSpm) activates the polyamine catabolic pathways in cultured tumor cells [175,176]. The over-expression of SAT1 in HEK293 cells led to reductions in Spm and Spd levels, disruption of cell cytoskeleton and adhesion, and reduced cell proliferation caused by cell cycle arrest at the G2 to M transition [15,177]. Growth arrest was in part due to severe DNA damage (single- and double-stranded breaks), which activated DNA damage/repair response proteins (e.g., ataxia telangiectasia mutated; ATM and ataxia telangiectasia and Rad3-related; ATR) effectively halting cell cycle progression [177]. Similarly, HEK293 cells combined with an adenovirus expression system demonstrated that enhanced expression of SAT1 leads to the depletion of Spm and Spd, growth arrest, and loss of cell viability [13,14]. The disruption of protein synthesis was another hallmark of increased SAT1 over-expression [13,14]. The reduction in polysomes and accumulation of monosomes, reduced eIF5A levels, and the onset of ERS/UPR underscored how the activation of this response can contribute to cell death in SAT1-over-expressing cells [39,99]. In both systems, the enhanced expression of SAT1 eventually led to mitochondrial alterations and apoptosis [14,168]. It is not completely clear what the contribution of each maladaptive pathway plays in inducing cell death in SAT1-expressing cells; however, all of the described cellular changes whether in isolation or in concert can be mediators of cell injury and tissue damage.

#### 4.2.3. The Role of Polyamine Catabolic Pathways in Normal Physiology

Although multiple studies have demonstrated the maladaptive role of elevated polyamine catabolism in disease conditions, there are few studies that address the importance of these pathways in normal physiological states. The harmful role of SMOX and its products in the mediation of acute and chronic tissue injury are well documented; however, the ablation of the Smox gene while protective in injuries does not seem to have an overt effect in normal physiological conditions other than the induction of a mild increase in tissue Spm and a significant reduction in tissue Spd levels. The baseline effect of SAT1 deficiency is more pronounced in that through a reduction in acetyl-coenzynme A (Acetyl-CoA) consumption for polyamine acetylation; it leads to increased availability of Acetyl-CoA for fatty acid synthesis, decreased oxidative phosphorylation, and ultimately an increase in total body fat accumulation over time [178,179].

Although individual ablation of either SAT1 and SMOX produces a mild phenotype, the combined deletion of both genes creates a severe condition in the organism. Examination of polyamine catabolism’s role in allergic airway inflammation (AAI—a model of asthma) in mice revealed that reductions in SAT1 or SMOX levels increase the severity of AAI [180]. In these studies, impeding SAT1 and SMOX expression by a combination of Berenil (diminazene aceturate) and MDL72527 also increased the severity of AAI providing additional evidence that the inhibition of both genes produces detrimental outcomes [180]. The physiological importance of polyamine catabolism is further confirmed through studies that utilized deletions in both genes to generate *Smox*/*Sat1*-dKO mice [84]. These animals, while normal at birth, went on to develop progressive cerebellar damage and ataxia. Cerebellar injury was associated with Purkinje cell loss and gliosis, leading to neuroinflammation and white matter demyelination [84]. The onset of tissue damage in *Smox/Sat1*-dKO mice was restricted to the cerebellum while other tissues and organs such as the spinal cord, lung, liver, and kidney were not affected [84]. These results led to the conclusion that the cerebellar injury was not solely dependent on changes in polyamine levels as it was highly selective [84]. The studies outlined in this section point out the important role of polyamine catabolic pathway in the maintenance of physiological homeostasis.

## 5. Conclusions

Polyamines (Figure 1A) are important regulators of a variety of molecular interactions and biological processes [1,2]. The intracellular content of these molecules is tightly regulated by an elaborate network of mechanisms including their import, export, *de novo* biosynthesis, and degradation (Figure 1B). These mechanisms constitute an exquisite control network that allows the cells and tissues to regulate the levels of polyamines within a physiological range in a manner that quickly responds to the prevailing conditions. The strict control of polyamine levels is necessary, as imbalances in their cellular pools can have many detrimental outcomes. For example, alterations in ODC and SMS functions can cause abnormalities in physical and mental development [137,156,157,159,160,161]. Potential alteration in lysosomal transport of polyamines caused by mutations in ATP13A2 are also associated with a form of juvenile onset Parkinsonism [55]. The depletion of polyamines can adversely affect molecular structures and interactions, while their elevated levels can also lead to chronic derangements that are harmful [1,2].

Likewise, polyamine metabolism can be a double-edged sword. The role of enhanced polyamine synthesis (ODC induction) in the prevention of tissue injury (e.g., cerebral IRI) [152,153,154,155], the development of neoplasia (e.g., MYC amplified tumors) [140,141], as well as developmental abnormalities (e.g., Bachmann–Bupp and Robinson–Snyder Syndromes) [156,161] has been documented. Increased polyamine catabolism has also been implicated in the mediation of acute (e.g., acute kidney injury, cerebral ischemic and traumatic injuries, and hepatic injuries) and chronic tissue injuries and the development of cancer (*H. pylori-* and *B. fragilis*-induced inflammation and GI tract cancers) [36,40,45,163,165,174]. Although not as prevalent, the adverse effects of disruptions in polyamine catabolism have also been documented in experimental settings [84,178,179,180]. These range from the propensity to develop late onset obesity to increased severity of AAI to the development of severe cerebellar damage and ataxia [84,178,179,180].

The cogent point of what has been discovered about polyamines and their catabolism is that they have important biological functions and that imbalance in their homeostasis and regulatory pathways can have adverse effects. In conclusion, as ongoing studies increase our knowledge of the biological significance of polyamines and their metabolism, our enhanced understanding of how to utilize polyamine and manipulate their metabolism will enable the development of new interventional and treatment modalities that can be used in a variety of diseases and genetic conditions.

## Figures and Tables

**Figure 1 medsci-10-00038-f001:**
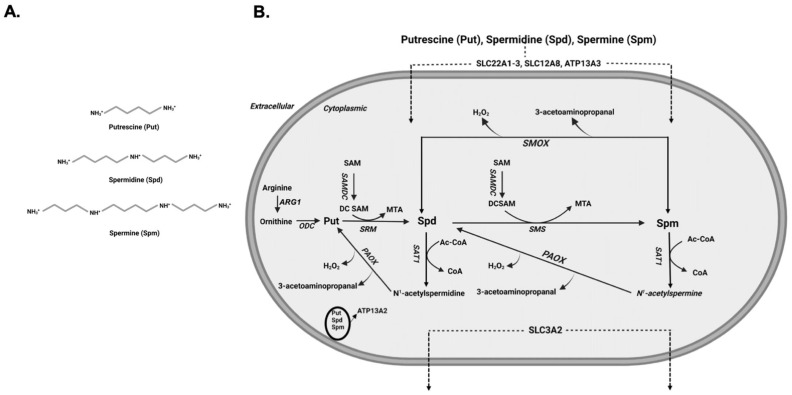
**Polyamine molecular structure and polyamine regulation.** (**A**) Chemical structure of Put, Spd and Spm. In physiological pH all three molecules are positively charged. (**B**) Schematic depiction of polyamine metabolic pathway and polyamine transporters. Arginase 1 (ARG1), Ornithine decarboxylase (ODC), SAM decarboxylase (SAMDC), Spermidine synthase (SRM), Spermine synthase (SMS), Spermidine/Spermine N1-acetyltransferase (SAT1), acetylpolyamine oxidase (PAOX) and Spermine oxidase (SMOX).

## Data Availability

Not applicable.

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
