# Peer review of "Polyamines and Their Metabolism: From the Maintenance of Physiological Homeostasis to the Mediation of Disease"

_medsci, 2022, doi:10.3390/medsci10030038_

Round 1

Reviewer 1 Report

Excellent summary of the different properties of polyamines. Congratulations for the work done

Author Response

We thank the reviewer for their kind comments and hope that this manuscript can be a useful reference for those interested in polyamines, their metabolism and their biology.

Reviewer 2 Report

The manuscript is well organized and covers a comprehensive area of polyamine research. I recommend publishing in Medical Science.

Minor points

1. Page 2, section 2.1. The font of the body needs to be changed to normal from italic.

2. Page 8, line 351. Please confirm Robinson-Snyder syndrome is Snyder-Robinson syndrome (SRS)?

3. Page 9, line 422. EIF5A might be eIF5A?

Author Response

We thank the reviewer for their kind review and informing us of the needed changes. As per reviewers suggestions the following corrections have been made. These correction are in blue font in the updated version of our manuscript.

1. Page 2, section 2.1. The font of the body needs to be changed to normal from italic.

The font has been converted from Italic to normal.

2. Page 8, line 351. Please confirm Robinson-Snyder syndrome is Snyder-Robinson syndrome (SRS)?

Robinson-Snyder syndrome has been corrected to Snyder-Robinson syndrome

3. Page 9, line 422. EIF5A might be eIF5A?

EIF5A has been changed to eIF5A

Reviewer 3 Report

The review summarizes recent knowledge about polyamines and their metabolism. The manuscript needs minor revision before its acceptance for publication.

Abstract

Lines 14-15, the sentence should be rephrased for clarity

Introduction

Lines 59-74, why in italic?

Line 69, G2/M should be explained

Lines 330-338, differences in study results; the differences in the study outcomes are almost contradictory. Can this simply be explained by study designs and animals used?

References

The references are not yet in the format prescribed in the instructions to authors: author names: A.W.; please only indicate journal volumes, but not issues

Author Response

We thank the reviewer for their kind comments. We have made the corrections and modifications asked by the reviewer. The modifications in the text of the manuscript are in blue.

Abstract

Lines 14-15, the sentence should be rephrased for clarity.

We have modified the sentence as follows: "This knowledge has been used for the development of novel compounds for research and medical applications." 

Introduction

Lines 59-74, why in italic?

The text has been converted to normal.

Line 69, G2/M should be explained. An explanation for G2/M transition phase in cell cycle has been added. "...G2/M transition phase, the transition between the second growth phase (growth phase after S phase) and mitosis in the cell cycle."

Lines 330-338, differences in study results; the differences in the study outcomes are almost contradictory. Can this simply be explained by study designs and animals used?

We agree with the reviewer that the results in reference 138 vs. references 152-155 are contradictory. To address this point, we have modified this section and excluded the conclusions made in reference 138. In the current manuscript, reference 138 is only used as evidence for the induction of ODC in cerebral IRI, while multiple other references (succeeding articles) that suggest a protective role for ODC in cerebral IRI are emphasized. The section now reads as follows: "The expression and activity of ODC increases following acute ischemic injuries in the targeted organs [138, 139]. The role of ODC in the mediation of tissue injury was most closely studied in cerebral IRI in rats. ODC-expressing transgenic rats were protected against cerebral ischemia, while the inhibition or knockdown of ODC exacerbated the severity of injury in multiple studies indicating a protective role for ODC [152-155]. These results suggest that ODC plays a protective role in cerebral IRI."  

References

The references are not yet in the format prescribed in the instructions to authors: author names: A.W.; please only indicate journal volumes, but not issues.

All references now use the ACS format. All citations throughout the text of the manuscript are now in brackets [ ]; and as per reviewer's suggestion, the issues have been removed from the listed references.